# Influence of Preparation Technology on Microstructural and Magnetic Properties of Fe_2_MnSi and Fe_2_MnAl Heusler Alloys

**DOI:** 10.3390/ma12050710

**Published:** 2019-02-28

**Authors:** Yvonna Jirásková, Jiří Buršík, Dušan Janičkovič, Ondřej Životský

**Affiliations:** 1CEITEC IPM, Institute of Physics of Materials, Academy of Sciences of the Czech Republic, Žižkova 22, 616 62 Brno, Czech Republic; jirasko@ipm.cz (Y.J.); bursik@ipm.cz (J.B.); 2Institute of Physics, Slovak Academy of Sciences, Dúbravská cesta 9, 845 11 Bratislava, Slovakia; Dusan.Janickovic@savba.sk; 3Department of Physics, VŠB-Technical University of Ostrava, 17 listopadu 15/2172, 708 33 Ostrava-Poruba, Czech Republic

**Keywords:** Heusler alloys, arc melting, planar flow casting, L2_1_ phase, microstructure, Curie temperature, temperature of re-orientation, hyperfine parameters

## Abstract

Microstructural and magnetic properties of the X_2_YZ, namely Fe_2_MnSi and Fe_2_MnAl, Heusler alloys have been studied from the viewpoint of technology for their production and for the Z element effect. First, arc melting was applied to produce button-type ingots from which samples in a form of 500 µm thick discs were cut. Second, planar flow casting technology yielded samples in a ribbon-form 2 mm wide and 20 μm thick. The checked area chemical compositions were in agreement with the nominal ones. Nevertheless, the darker square objects and smaller bright objects observed at the wheel side of the Fe_2_MnSi ribbon sample yielded higher Mn content at the expense of Fe. The X-ray diffraction patterns of all samples have indicated L2_1_ structure with lattice parameters, 0.567 (1) nm for Fe_2_MnSi and 0.584 (1) nm for Fe_2_MnAl, being within an experimental error independent of production technology. On the other hand, the technology has markedly influenced the microstructure clearly pointing to the larger size of grains and grain boundaries in the disc samples. From the magnetic viewpoint, both alloys are paramagnetic at room temperature without visible influence of their production. On the contrary, the low-temperature behavior of the microscopic hyperfine parameters and the macroscopic magnetic parameters exhibits differences affected by both chemical composition and microstructure.

## 1. Introduction

The Heusler compounds of general chemical formula XYZ (half-Heusler) or X_2_YZ (full-Heusler) have since the discovery by F. Heusler in 1903 an important position among materials. X and Y are transition metals and Z is in the p-block consisting of five columns 13–17. Many of these compounds exhibit properties relevant to spintronics, such as magnetoresistance, variations of the Hall effect, ferro-, antiferro-, and ferrimagnetism, half- and semimetallicity, and topological band structure. Their magnetism results from a double-exchange mechanism between neighboring magnetic ions. A chemical element present in the first Heusler alloy and being frequently used also in the next compositions is manganese. It is used, e.g., in the ferromagnetic shape memory Heusler materials [1]. Development of magnetic Heusler alloys based on Co and Mn with high Curie temperature and high magnetoresistance effect, specifically designed for spintronic applications, meant recently the huge number of investigations, e.g., References [2,3,4,5]. Similarly, Fe_2_YSi (where Y = Cr, Mn, Co, Ni) alloys were studied both experimentally and theoretically [6,7] and various other compositions were used as thin films in multilayer systems [8,9]. 

Besides ternary Heusler compounds, a large effort has been devoted also to quaternary alloys of the type X_2_Y_1−*x*_Y’*_x_*Z or X_2_YZ_1−*x*_Z’*_x_*. Belkhouane and co-workers have studied theoretically the ternary Fe_2_MnSi, Fe_2_MnAl alloys and quaternary composition Fe_2_MnSi_0.5_Al_0.5_ [10]. Their results concerning the calculated lattice constants and spin magnetic moments were found to be in good agreement with the available experimental and theoretical data. The half-metallic behavior was identified at Fe_2_MnAl while nearly half-metallic behavior with a small spin-down electronic density of states at Fermi level was indicated at Fe_2_MnSi_0.5_Al_0.5_. Another example of the quaternary Heusler alloy is a system Co_2_Cr_1−*x*_Fe*_x_*Al studied by Fecher et al. [11]. 

The Heusler alloys are produced by various synthesis procedures as sputtering or evaporation resulting in films, e.g., References [7,8,12], arc or induction melting yielding bulk materials [13,14,15], mechanical alloying which results in powders [16], melt spinning gives samples in a form of ribbons, [14]. All these technologies influence the structure, morphology, thereby physical properties of final products.

The main objective of the present research was to verify the production of both Heusler Fe_2_MnSi and Fe_2_MnAl alloys by two substantially different technological procedures, namely arc melting and planar flow casting, to extend the previous results [14] and to compare the influence of the *p*-element (Si, Al). The Heusler L2_1_ structure consists of four fcc sublattices mutually shifted by a vector (1,1,1) *a*/4, where *a* denotes the fcc lattice parameter. The unit cell of Fe_2_Mn(Si, Al) is depicted in Figure 1. The sites (sublattices) are denoted by A (red), B (orange), C (blue), and D (green) whereas the A and C sites are equivalent and occupied by Fe in a well-ordered stoichiometric structure, B sites are occupied by Mn and D sites by Si or Al atoms.

## 2. Materials and Methods 

High purity elements; 99.95% Fe, 99.9% Mn, 99.9%, Si, and 99.95% Al were used for the production of both alloys, Fe_2_MnSi and Fe_2_MnAl. A conventional arc melting (AM) procedure done under an argon atmosphere using a Compact Arc Melter MAM-1 furnace (Edmund Bühler GmbH, Bodelshausen, Germany) resulted in button-type ingots (Figure 2a). They were re-melted four times for homogeneity reasons (the weight loss was close to 1%). The other procedure, known from the production of amorphous and/or nanocrystalline ribbons, was planar flow casting (PFC). The initially-prepared ingot was melted in a quartz crucible and then the molten alloy was ejected through the nozzle by supplying a high purity argon gas to the crucible onto rotating Cu wheel at air. The polycrystalline ribbons (R) of the Fe_2_MnSi (denoted RS) and of the Fe_2_MnAl (denoted RA) were 20 μm thick and 2 mm wide (Figure 2b). As usual, they featured structurally different surfaces on the wheel and air sides. The AM ingots were cut into discs about 500 μm thick using spark erosion in deionized water and subsequently, their surfaces were grinded and polished to remove oxides and to form a flat surface. These samples of Fe_2_MnSi and Fe_2_MnAl are denoted DS and DA, respectively. In addition, the selected samples were polished using a Vibromet polishing machine for 24 h to guarantee the good surface smoothness. Small pieces of samples, about 3 mm in diameter, were prepared also for magnetic measurements. The PFC samples of both alloys were used in their as-prepared state. No additional mechanical surface treatment was possible because of their high brittleness.

In order to evaluate both technological procedures and both chemical compositions in detail, several experimental methods were applied. 

A TESCAN LYRA 3XMU FEG/SEM scanning electron microscope (SEM, Brno, Czech Republic) working at accelerating voltage of 20 kV and equipped with an XMax80 Oxford Instruments detector (UK) for energy dispersive X-ray (EDX) analysis were used for the chemical surface morphology and composition measurements. 

X-ray diffraction (XRD) measurements were done using an X’PERT PRO powder diffractometer (Malvern Panalytical, UK) with Co-Kα radiation (λ = 0.17902 nm) at room temperature (RT) in the range of 2θ from 25° to 135° with steps of 0.008° and 500 s per degree. The patterns were analyzed using HighScore Plus software (Version 4.8.0) including Rietveld structure refinement method [17] and by applying the external ICSD database (FIZ Karlsruhe, Karlsruhe, Germany) of inorganic and related structures [18]. 

Mössbauer spectroscopy (MS) measurements at room temperature were done using a ^57^Co (Rh) source. Transmission Mössbauer spectroscopy (TMS) using home made equipment was applied for the ribbon-type samples (RS, RA) due to their favorable thickness allowing also Mössbauer measurements below RT. The backscattering Mössbauer spectroscopy (γ-BMS) of the penetration depth about 25 μm at RT only was used for the disc-type samples (DS, DA). The velocity scales in both measuring geometries were calibrated using α-Fe at room temperature. All spectra were evaluated within the transmission integral approach using the CONFIT program [19]. 

Magnetic properties at elevated temperatures (293–573 K) and in an applied external field of ±1600 kA/m (±2 T) were measured using a vibrating sample magnetometer (VSM) EV9 (Microsense, MA, USA). A physical property measurement system (PPMS, Model P935A, San Diego, CA, USA) quantum design was applied for measurements of hysteresis loops with maximal magnetic field of ±4000 kA/m (±5 T) and zero-field-cooled (ZFC) and field-cooled (FC) curves in the magnetic field of 8 kA/m and in the temperature range (2–293 K). 

## 3. Results and Discussion

### 3.1. SEM—Morphology and Chemical Composition

The surface morphology of the arc-melted samples (left-DS, right-DA) is seen in Figure 3a. The wheel-side morphologies of the ribbon-type samples of both compositions (RS, RA) are shown in Figure 3b and morphology details in Figure 3c. The same way is used for morphology imaging of the air-sides of both ribbon-type samples in Figure 3d,e. The EDX chemical analysis taken from an area of about (1 × 1) mm^2^ resulted in good agreement with nominal composition mainly at the AM disc samples (DS, DA) where the homogeneity of the chemical composition was also satisfactory. This is reflected in a small dispersion of measured values in Table 1. Slightly different chemical compositions between the wheel and air sides were found at ribbon samples (RS, RA). The largest local inhomogeneities of chemical composition were detected at the wheel side of the Fe_2_MnSi sample (RS) (Figure 3b, left panel). Point EDX analyses split into three groups: darker square objects yielded higher content of Mn at the expense of Fe (38.3 ± 0.4 at.% Fe, 33.9 ± 0.6 at.% Mn and 27.8 ± 0.1 at.% Si), smaller bright objects yielded even higher content of Mn (in at.%: 26 Fe, 40 Mn, 34 Si), and the majority gray matrix yielded 45.3 ± 0.1 Fe, 27.3 ± 0.1 Mn, 27.4 ± 0.1 Si (at.%) close to the nominal composition. All formations are well visible in Figure 3c, left panel. Contrary to this situation, no similar inhomogeneities were observed at the wheel side of the RA sample (Figure 3c,d, right panel) even if the surface morphology was very similar. The large grains of dimensions ranging between 250 μm and 300 μm were observed at samples prepared by AM (DS, DA—Figure 3a) while the grains of both samples prepared by PFC were at least two orders of magnitude smaller in agreement with the higher cooling rate used at this technology. They are in detail seen in Figure 3e. 

### 3.2. XRD—Chemical and Phase Structure

The results of XRD measurements are seen in Figure 4 for both compositions and both technologies. Details of patterns in the 2θ range from 30° to 40° should better promote the weak (111) and (200) peaks. The peak (111) for Heusler alloy of general formula X_2_YZ reflects the ordering of Y (Mn) and Z (Si, Al) atoms and (200) corresponds to the superlattice reflections of X (Fe) atoms. The peak (220) is principal reflection independent on order [20]. 

In the Fe_2_MnSi diffractogram of the DS sample (Figure 4a) the lack of the (111) peak indicates a large amount of disordered occupation between Mn and Si, while the existence of (200) superlattice peak refers to the ordering of the Fe sublattice. At the air side of the RS sample, both peaks can be identified contrary to the wheel side where only (111) peak is visible. The Rietveld analysis for the DS and RS (air side) was done using the ICSD data sheet 659018 (Fe/Mn/Si = 2/1/1) and by the sheets 57284 (Fe/Mn/Si = 3/3/2) and 186061 (Fe/Mn/Si = 2/1/1) for the RS wheel side. This result is in agreement with inhomogeneities observed by the EDX analysis in dark and bright objects where the higher Mn content at the expense of Fe was detected. The line-asymmetries seen on the left side of the (200) and (400) peaks at the DS sample are caused by a different stoichiometry of crystals and size of coherent domains. In the Fe_2_MnAl diffractograms (Figure 4b) the main reflections are (220) and (400); the even (200) and odd (111) superstructure reflections are visible in both DA and RA samples. Moreover, both sides of ribbons are identical as it was also obtained from both area and point EDX analysis. The Rietveld analysis resulted in lattice parameters, a, summarized in Table 2, along with some of the other theoretical [10,21,22,23] and experimental [24,25,26] results obtained for bulk samples. It is evident that the experimentally determined lattice parameters for both Heusler compositions are slightly higher compared to theoretical ones obtained for stoichiometric compositions and well ordered, defect-free structures. 

### 3.3. MS—Mössbauer Spectroscopy

Mössbauer spectra in Figure 5 measured at room temperature in backscattering (DS, DA) and transmission (RS, RA) geometries confirm the paramagnetic state of both Fe_2_MnSi (Figure 5a) and Fe_2_MnAl (Figure 5b) alloys. All RT spectra were analyzed by a dominant single-line and minor double-line components revealing no substantial differences in hyperfine parameters. The low-temperature measurements of the ribbon-type samples reflect differences between the RS and RA. The first weak magnetic dipole interactions at the Fe_2_MnSi (RS) alloy have appeared in the Mössbauer spectrum measured at 200 K. This is approximately consistent with a Curie temperature of 214 K presented, e.g., in Reference [27]. The measurements at lower temperatures of the 150 K, 60 K, and 6 K clearly document the ferromagnetic behavior of this alloy. In addition to the slightly split six-line components revealing weak hyperfine inductions, also the sextuplets corresponding to larger values of hyperfine induction are detected. Therefore, the spectra were analyzed using several sextuplets revealing different values of hyperfine induction and small values of quadruple splitting (~0.02 mm/s). The hyperfine inductions can be divided into three intervals; between 0 and 10 T, 11–20 T, and 21–30 T. Their mean values (*B1_m_, B2_m_, B3_m_*) and fractional intensities (*A1*, *A2*, *A3*) are presented in Table 3.

The values of hyperfine induction are influenced by the type and number of atoms in the nearest-neighborhood of resonating Fe atom. To explain the individual values of hyperfine induction and their relative intensities in more details the experiments leading to the elimination of disorder of the L2_1_ structure had to be done. The comparable spectra were measured at 5.5 K and 78 K at the powdered Fe_2_MnSi sample yielding approximately similar mean values of hyperfine induction corresponding to present values *B1_m_* and *B2_m_* (Table 3) but a higher value compared to *B3_m_* [28].

The Mössbauer spectra of the Fe_2_MnAl (RA) sample at low temperatures are not so well split into sextuplets as was observed at the RS samples. At low temperatures, they exhibit broad lines representing both electric quadrupole and magnetic dipole hyperfine interactions. A two-pattern fit of the experimental data using Gaussian distributions of quadrupole split doublets P(*Δ*) and magnetically split sextuplets P(*B*) was used to analyze the spectra measured between the 140 K and 5 K. The ferromagnetic component is in the spectrum at 5 K represented by the mean value of hyperfine induction *B_m_* = (14.4 ± 0.5) T and intensity *A* about 95 %. Both values decrease with increasing temperature to *B_m_* = (8.3 ± 0.4) T, *A* = 51 % at 50 K, and to *B_m_* = (8.9 ± 0.3) T, *A* = 3.4 % at 140 K. This diminishing of magnetic dipole interactions with increasing temperature is connected with a transition of the RA sample into the paramagnetic state. The Curie temperature can be supposed slightly above 140 K because no sextuplets appeared at 200 K. A Curie temperature of 150 K was determined at the melt spun ribbon-type Fe_2_MnAl sample by magnetic measurements in Reference [29], which supports the present result. Nevertheless, the authors investigating the Fe_2+*x*_Mn_1−*x*_Al (*x* = −0.1, 0.0, and 0.1) samples prepared by arc melting declare that the main ordered L2_1_ phase in these alloys order ferromagnetically with *T_C_* ~ 300 K for *x* = 0.0 and −0.1, and 380 K for *x* = 0.1 [30]. Their Mössbauer measurements in transmission geometry at 80 K were done at the powder samples prepared from the central part of the bulk samples. The measurements yielded spectra formed by broadened six-lines for all values of *x* documented the ferromagnetic order of the samples. Two sextets and a singlet were obtained by Mössbauer measurements of the Fe_2_MnAl sample prepared by ball milling at 300 K and only the paramagnetic singlet has appeared at 373 K [31]. It seems that the Fe_2_MnAl alloy prepared by severe deformation (mechanical alloying, ball milling) is sensitive to induced stresses evoking ferromagnetic order at RT. Another reason could be contamination of samples by tools used during the powder preparation. 

### 3.4. Magnetic Properties

Paramagnetic behavior of all as-prepared Fe_2_MnSi (DS, RS) and Fe_2_MnAl (DA, RA) samples at room temperature, resulting in Mössbauer spectroscopy in the paramagnetic singlet, is also confirmed by macroscopic magnetic measurements. It is documented by zero magnetization at RT at zero-field-cooled (ZFC) and field-cooled (FC) curves in Figure 6 and by non-saturated hysteresis loops at elevated temperatures (373 K, 573 K) shown in Figure 7 and hysteresis loops even in a high magnetic field of 5 T depicted in Figure 8. Nevertheless, the weak magnetization reversal in the vicinity of low magnetic fields was observed at all samples. It is documented by the hysteresis loop of the DA sample depicted in inset of Figure 8c. This indicates a small contribution of residual ferromagnetism manifested by non-zero coercivity.

Both magnetic contributions, paramagnetic and weak ferromagnetic, are also observed at hysteresis loops measured from the RT (Figure 8) up to 573 K (Figure 7). These measurements allow determination of the Curie temperature *T_C_* using the Curie-Weiss law expressed by:1/χ = (*T* − *T_C_*)/*C*(1)

Here χ is the magnetic susceptibility obtained from the paramagnetic part of a hysteresis loop measured at temperature *T* and *C* is the Curie constant. Analysis of a linear dependence *χ*^−1^ vs *T* for all samples provides the *T_C_* and *C* parameters shown in Table 4. The Curie temperature*, T_C,_* of the RS sample is shifted towards the room temperature and the value of Curie constant is approximately 1.45 times lower than that for the DS sample. Independently on the Curie-Weiss law, the Curie temperature was also determined from the ZFC-FC curves (Figure 5) using one tangent and/or two tangent method [32]. This resulted in *T_C_* ~216 K for the DS sample in good agreement with the values presented in Table 4 and comparable with 220 K presented for ingot-type samples, e.g., in Reference [33]. The Curie temperature of the RS sample determined by the tangent methods was about 22 K lower compared to that in Table 4.

The Curie temperature of the Fe_2_MnAl (DA) sample determined by both procedures (Curie-Weiss law and tangent methods) resulted in a value of ~137 K that is slightly lower compared to 150 K presented, e.g., in Reference [29]. Two critical points, ~170 K and ~260 K, exist on the ZFC-FC curves of the RS sample. In this case, an application of tangent methods is not suitable. The Curie-Weiss law yields the *T_C_* value of 263 K (Table 4) that is close to the second mentioned critical point and to values presented by other authors [30,31] while the lower value of 170 K is in better agreement with the value estimated from the present Mössbauer measurements.

The formation of ferromagnetic ordering with decreasing temperature, consistent with MS measurements, is seen for all samples on ZFC curves in Figure 5. The magnetization achieves maximum at a temperature called the temperature of re-orientation *T_R_*. The *T_R_* at the Fe_2_MnSi samples is practically independent on their production and the value of about 69 K is in good agreement with the literature [27]. In the case of Fe_2_MnAl, the value of 42 K was obtained for the DA sample while for the RA sample this value is closer to those obtained for the Fe_2_MnSi samples (65 K). A reason for this difference is not clear at present. The temperature of 51 K was presented, e.g., in References [25,29]. The magnetic behavior of all samples below RT is also manifested by changes of the hysteresis loops in Figure 6 from which the increasing values of magnetization in the field 5 T (*M*5), remnant magnetization (*M_r_*)*,* and coercivity (*H_c_*) were determined. The magnetizations do not saturate up to 5 T, suggesting a disordered magnetic system which seems to be higher at the Fe_2_MnAl samples. It is consistent with the Mössbauer result (RA in Figure 5). Simultaneously, the observed marked increase of the structurally sensitive characteristics *M_r_* and *H_c_* at 2 K (Table 4) indicates an inhomogeneous magnetic system in both Heusler alloys as well. Similar conclusions were presented in References [27,29] suggesting a competition between ferromagnetic and antiferromagnetic interactions. According to Figure 1, the Mn atoms preferentially occupy the *B* sites and the Si or Al atoms occupy the *D* sites. The other sites, *A* and *C*, are equivalent and they are occupied by Fe atoms. However, it has been reported that such site occupation is often distorted, and Fe atoms occupy also the *B* sites [34]. In such a way a random substitution of Fe for Mn can contribute to the large distribution of ferromagnetic interaction energies resulting in a broader transition at the *T_C_*. As the curves in Figure 5 document, it is more significant at the alloys prepared by the planar flow casting technology (RS, RA). 

Spontaneous magnetization obtained by extrapolating *M*-*H* curves to zero field estimated from 2 K hysteresis loops is for both Fe_2_MnSi samples about 53 Am^2^/kg (1.85 μ*_B_*/f.u.), while the coercive fields are 8.61 kA/m for DS and 16.02 kA/m for RS. The present value of spontaneous magnetization is smaller than experimentally obtained 74 Am^2^/kg (2.6 μ*_B_*/f.u.) [26], 61 Am^2^/kg (2.1 μ*_B_*/f.u.) [35], and also less than theoretically calculated ~3 μ*_B_*/f.u. [10]. The same procedure used for the Fe_2_MnAl composition yielded spontaneous magnetization 28.1 Am^2^/kg (0.97 μ*_B_*/f.u_._) and 73.13 kA/m for DA and 39.5 Am^2^/kg (1.37 μ*_B_*/f.u) and 32.12 kA/m for RA samples. The spontaneous magnetization 34 Am^2^/kg (1.18 μ*_B_*/f.u_._) was observed by Kourov et al. [26], 38 Am^2^/kg (1.32 μ*_B_*/f.u_._) by Liu et al. [29], and 1.58 μ*_B_*/f.u. was presented by Buschow et al. [25]. All these values obtained experimentally are less than the calculated total magnetic moment 2.00 μ*_B_*/f.u. [10,36]. 

## 4. Conclusions

The paper summarizes the systematic magnetic study of full Heusler Fe_2_MnZ alloys crystallizing in L2_1_ structure prepared in two compositions and using two different technological procedures. The composition differs in Z atom being either Si or Al. The effect of producing technology consists of different cooling rates of a melt influencing also a form of the final product. Bulk ingots were produced by arc melting (AM) and ribbons of 20 μm thick by planar flow casting (PFC). Both technological procedures resulted in the Heusler compound as documented by XRD. The technology has influenced the grain size being 250–300 μm in bulk samples and two orders smaller in ribbons. The chemical composition was homogenous and in good agreement with the nominal composition at the AM disc samples, slight deviations were obtained at PFC Fe_2_MnSi ribbon. 

From the magnetic viewpoint, both alloys are paramagnetic at room temperature and they transform into a ferromagnetic state at decreasing temperature. It is accompanied by changes in macroscopic as well as microscopic magnetic characteristics manifested by changes in hysteresis loops and by splitting the Mössbauer spectra into magnetic subcomponents. No saturation magnetization at low temperatures is obtained up to an external field of 5 T manifesting magnetically inhomogeneous systems characterized by a competition of ferromagnetic and antiferromagnetic ordering. The comparison of studied samples from viewpoint of different technological procedures yields no substantial differences between the arc melted samples and those prepared using planar flow casting. Slight differences are observed from a viewpoint of chemical composition. It seems that the structural and magnetic properties of the Fe_2_MnAl alloy are more sensitive to atomic ordering in the Heusler structure.

## Figures and Tables

**Figure 1 materials-12-00710-f001:**
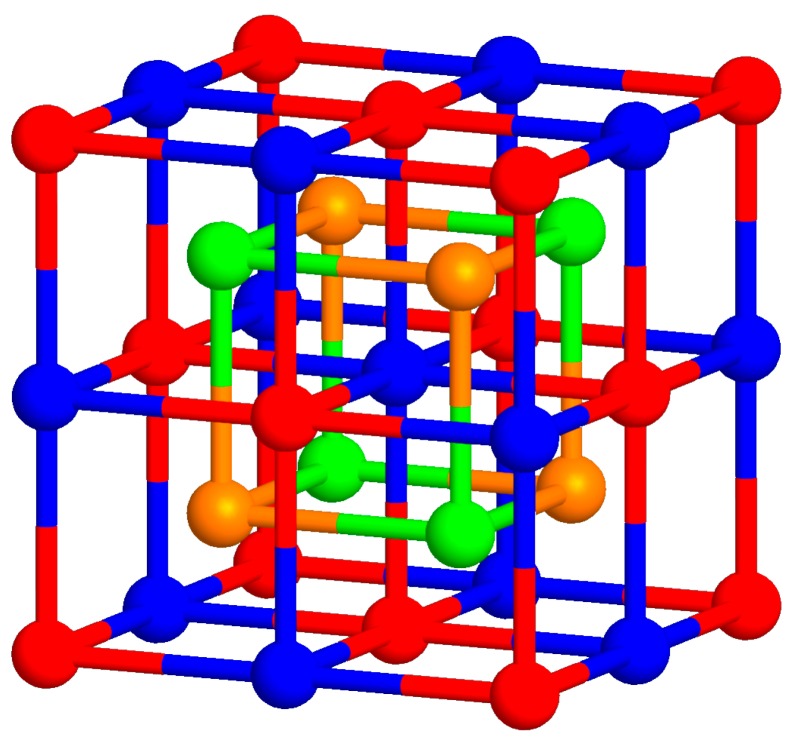
Unit cell of Heusler Fe_2_Mn(Si, Al) alloy. The sites are denoted by A (red), B (orange), C (blue), and D (green).

**Figure 2 materials-12-00710-f002:**
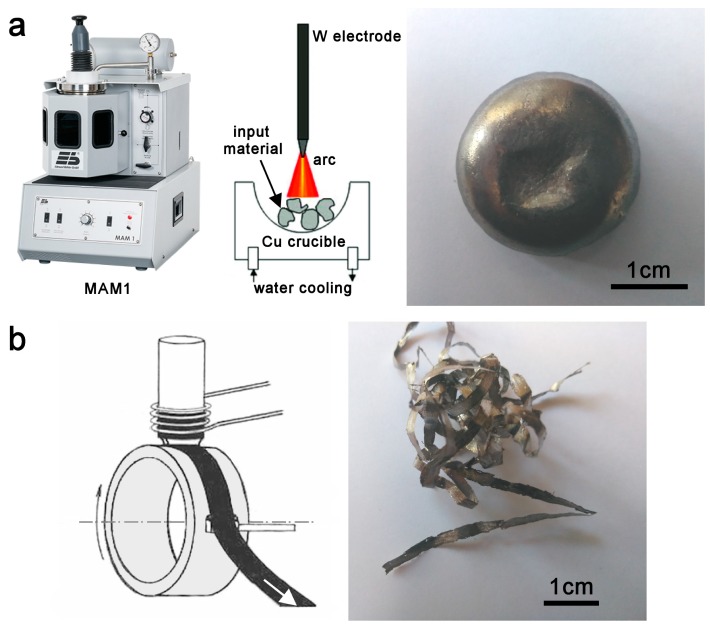
(**a**) MAM-1 furnace with a schematic representation of the melting chamber and button-type ingot; (**b**) schematical drawing of planar flow casting and ribbon-type sample.

**Figure 3 materials-12-00710-f003:**
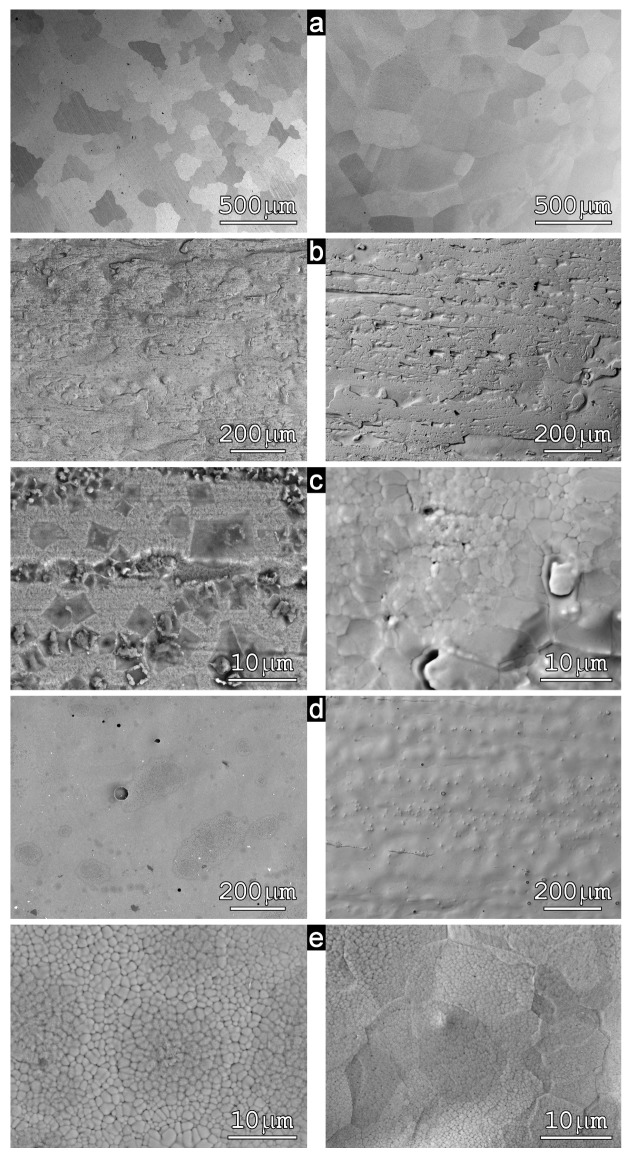
SEM micrographs of the Fe_2_MnSi (left panel) and Fe_2_MnAl (right panel) samples prepared by arc melting: DS, DA (**a**) and by planar flow casting-wheel side: RS, RA (**b**,**c**), air side: RS, RA (**d**,**e**). Details are described in the text.

**Figure 4 materials-12-00710-f004:**
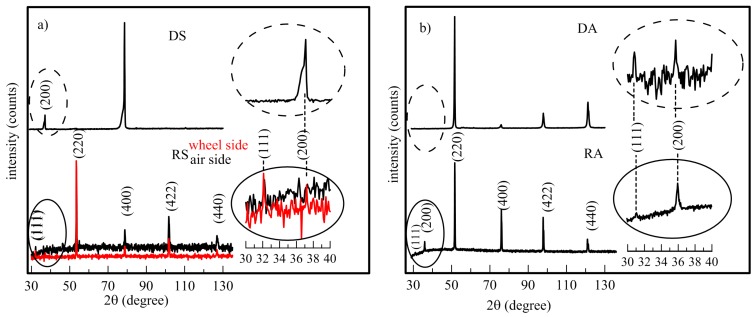
X-ray diffraction patterns for Fe_2_MnSi (**a**) and Fe_2_MnAl (**b**) Heusler alloys prepared by arc melting (DS, DA) and planar flow casting (RS, RA).

**Figure 5 materials-12-00710-f005:**
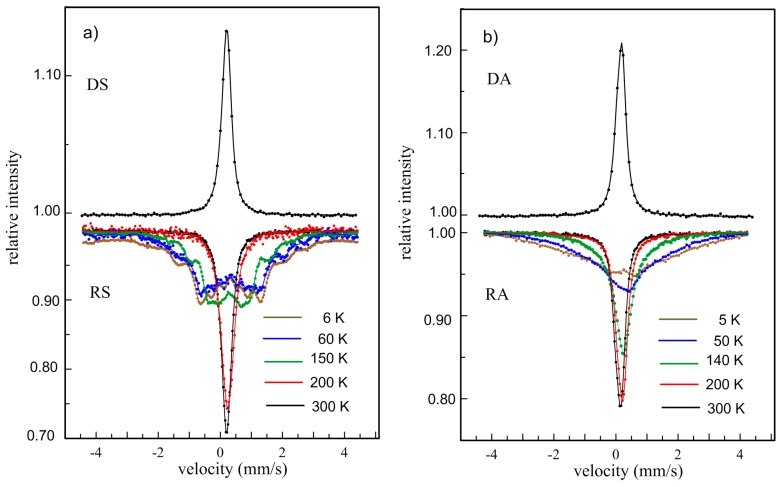
Room-temperature Mössbauer spectra of the Fe_2_MnSi (**a**) and Fe_2_MnAl (**b**) Heusler alloys prepared by arc melting (DS, DA) and planar flow casting (RS, RA) and low-temperature Mössbauer spectra for the RS and RA samples.

**Figure 6 materials-12-00710-f006:**
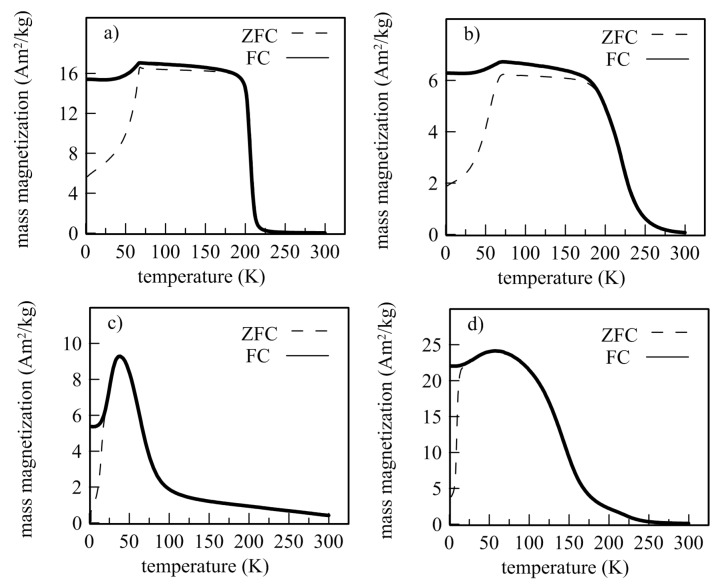
ZFC-FC curves of Fe_2_MnSi (**a**, DS; **b**, RS) and Fe_2_MnAl (**c**, DA; **d**, RA) alloys measured in magnetic field of 8 kA/m.

**Figure 7 materials-12-00710-f007:**
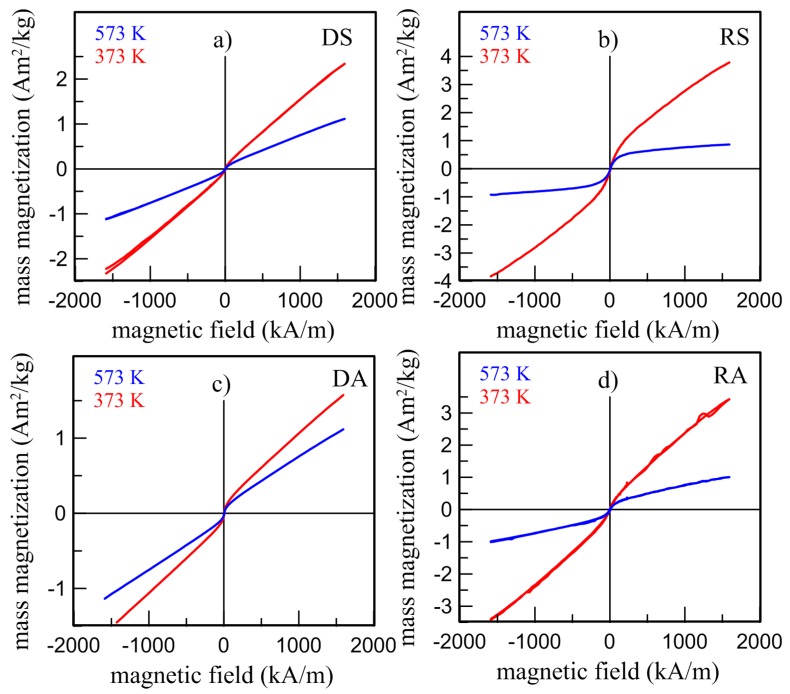
Hysteresis loops of Fe_2_MnSi (**a**, DS; **b**, RS: top panels) and Fe_2_MnAl (**c**, DA; **d**, RA: bottom panels) alloys measured at elevated temperatures *T* = 373 K, 573 K.

**Figure 8 materials-12-00710-f008:**
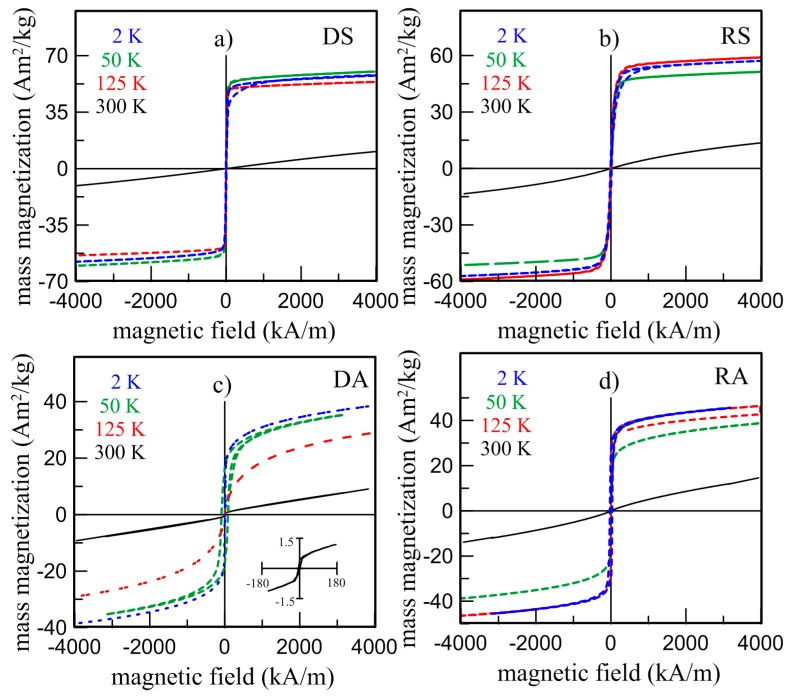
Hysteresis loops of Fe_2_MnSi (**a**, DS; **b**, RS: top panels) and Fe_2_MnAl (**c**, DA; **d**, RA: bottom panels) alloys measured at constant temperatures *T* = 2 K, 50 K, 125 K, and 300 K. The inset in bottom panel (**c**) presents detail of hysteresis loop at 300 K.

**Table 1 materials-12-00710-t001:** Chemical composition taken from an area of about (1 × 1) mm^2^ of the Fe_2_MnSi and Fe_2_MnAl samples prepared by arc melting (DS, DA) and planar flow casting (RS, RA).

Sample		Fe (at.%)	Mn (at.%)	Si/Al (at.%)
DS		48.30 ± 0.14	24.69 ± 0.09	27.01 ± 0.05
DA		49.95 ± 0.31	22.98 ± 0.12	27.07 ± 0.25
RS	air side	47.10 ± 0.19	27.62 ± 0.52	25.28 ± 0.36
	wheel side	43.36 ± 0.67	30.54 ± 0.19	26.10 ± 0.84
RA	air side	49.27 ± 0.16	24.51 ± 0.30	26.22 ± 0.45
	wheel side	48.36 ± 0.54	24.24 ± 0.17	27.40 ± 0.70

**Table 2 materials-12-00710-t002:** Measured lattice parameters, a (nm), for the Fe_2_MnSi (DS, RS) and Fe_2_MnAl (DA, RA) samples; theoretical data (theor.) and experimental data (exp.) for bulk samples of other authors are quoted for comparison.

Compound	Fe_2_MnSi			Fe_2_MnAl	
	DS	RS		DA	RA
		Air Side	Wheel Side		
present exp.	0.5685 (40)	0.5669 (1)	0.5667 (1)	0.5847 (3)	0.5668 (1)
[21] theor.	-			0.5683	
[10] theor.	0.5599			0.5680	
[24,25] exp.	0.5671 [24]			0.5816 [25]	
[26] exp.	0. 5671			0.5836	
[22] theor.	0.5601			-	
[23] theor.	-			0.5590	

**Table 3 materials-12-00710-t003:** Mean values of the hyperfine induction (*B_m_*) and fractional intensities (*A*) obtained for Fe_2_MnSi (RS) samples prepared by planar flow casting at low temperatures (see text). The *A_p_* corresponds to residual paramagnetic phase.

*T* (K)	*B1_m_* (T)	*A1* (%)	*B2_m_* (T)	*A2* (%)	*B3_m_* (T)	*A3* (%)	*A_p_*
150	8.015	34.8	12.991	5.0	22.411	10.0	50.2
60	6.977	71.3	11.268	14.0	24.465	14.7	0
6	6.561	65.3	11.865	20.0	24.302	14.7	0

**Table 4 materials-12-00710-t004:** Magnetic parameters of Fe_2_MnSi (DS, RS) and Fe_2_MnAl (DA, RA) alloys determined from hysteresis loops (magnetization at 5 T, *M*5; remnant magnetization, *M_r_*; coercivity, *H_c_*), by applying the Curie-Weiss law (Curie constant, C; Curie temperature, *T_C_*), and from a maximum on zero-field-cooled (ZFC) curve (temperature of re-orientation, *T_R_*).

Sample	*T* (K)	*M*5 (Am^2^/kg)	*M_r_* (Am^2^/kg)	*H_c_* (kA/m)	*C* (m^3^·K/kg)	*T_C_* (K)	*T_R_* (K)
DS	300	10.55	0.02	2.22	2.37 × 10^−4^	216	68.58
	125	53.77	2.45	1.64			
	50	60.15	3.54	1.70			
	2	57.83	11.22	8.61			
RS	300	13.55	0.01	1.13	1.64 × 10^−4^	272	69.02
	125	51.33	3.51	4.10			
	50	59.03	4.15	6.30			
	2	57.05	10.11	16.02			
DA	300	10.36	0.29	3.52	2.82 × 10^−4^	137	42.02
	125	29.02	0.64	3.32			
	50	38.57	1.54	1.04			
	2	39.53	19.12	73.13			
RA	300	14.39	0.02	1.63	1.60 × 10^−4^	263	65.25
	125	39.76	4.39	2.41			
	50	47.95	8.06	3.74			
	2	48.66	32.15	32.12			

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
