# Peer review of "Influence of Preparation Technology on Microstructural and Magnetic Properties of Fe2MnSi and Fe2MnAl Heusler Alloys"

_materials, 2019, doi:10.3390/ma12050710_

Reviewer 1 Report

Attached below.

Author Response

Dr. Yvonna Jiraskova

Institute of Physics of Materials,

Academy of Sciences of the Czech Republic,

Zizkova 22, 616 62 Brno

Czech Republic         

February 19, 2019

We thank Reviewer for his/her comments leading to improvement of our manuscript. We have accepted all of them.

All changes are marked by red colour in the revised version of the manuscript and corresponding answers and information are added here:

Reply to Reviewer 1

Reviewer

(1) In first paragraph, the authors stated that Heusler compounds are attractive from the viewpoint of spintronics. Some appropriate references should be cited.

Answer

Besides Ref. [2] additional three papers concerning applications of Heusler alloys in spintronics are added in the corrected manuscript as Refs. [3-5]:

   [3]            Block, T.; Wurmehl, S.; Felser, C.; Windeln, J. Powder magnetoresistance of Co2Cr0.6Fe0.4Al/Al2O3 powder compacts. Appl. Phys. Lett. 2006, 88, 202504, DOI: 10.1063/1.2200571.

   [4]            Bainsla, L.; Raja M.M.; Nigam, A.K.; Suresh, K.G. CoRuFeX (X = Si and Ge) Heusler alloys: High TC materials for spintronic applications. J. Alloys Comp. 2015, 651, 631-635, DOI: 10.1016/j.jallcom.2015.08.150.

   [5]            Mizukami, S.; Serga, A.A. Advancement in Heusler compouynds and other spintronics material designs and applications. J. Phys. D:Appl. Phys. 2015, 48, 160301, DOI: 10.1088/0022-3727/48/16/160301.

Reviewer

(2) In Method section, authors described the magnetization measurements at high temperature (293K - 573 K). However, the measurement results at high-temperature were not presented.

Answer

The hysteresis loop at 300 K was measured using PPMS within the low-temperature measurements. The hysteresis loops at elevated temperatures of 373 K and 573 K are similar to those measured at room temperature and they were measured by vibrating sample magnetometer in an applied field ± 2000 kA/m and used predominantly for the Curie temperature determination. Nevertheless we have accepted this note and added Fig. 7 where the hysteresis loops measured at 373 K and 573 K for all samples are depicted.

Figure 7. Hysteresis loops of Fe2MnSi (DS, RS: top panels) and Fe2MnAl (DA, RA: bottom panels) alloys measured at elevated temperatures T = 373 K, 573 K.

Reviewer

(3) The pictures of samples, particularly for ribbons, would be of interests for potential readers.

Aswer

Section 2 “Materials and Methods” was newly completed by inserted Fig. 2 showing schematically both arc melting and planar flow casting methods resulting in button- and ribbon-type samples, respectively.

Figure 2. a) MAM-1 furnace with schematic representation of melting chamber and button-type ingot; b) schematical drawing of planar flow casting and ribbon-type sample.

Reviewer

(4) XRD patterns for the air side of the RS samples that contains secondary phases should be presented.

Answer

Both XRD patterns (wheel-side: red line; air-side: black line) of the RS sample were presented in figure 3 (figure 4 in the corrected manuscript). The amount of secondary phase obtained by Rietveld analysis from the wheel-side was small (~ 3-5 %) and therefore it is not seen in the scale of the XRD pattern in figure 4.  

Reviewer

(5) In Fig. 3(a), diffraction peaks for DS sample have shoulder at lower angles. The origin of this shoulder should be discussed.

Answer

This comment is newly related to figure 4 of the corrected manuscript. The samples were studied in the as-prepared disordered state and therefore not well defined Heusler structure can be expected. The following sentence was added into the text of subsect. 3.2:

The line-asymmetries seen on the left-side of the (200) and (400) peaks at the DS sample are caused by a different stoichiometry of crystals and size of coherent domains.

Reviewer 2 Report

In general, the paper is well prepared and presented but it need some corrections and comments. The microstructere and phase analyses need improvement.

page 4, line 128 - fig. 2. - the SEM micrographs should be evaluated, both - on the pictures - there are not any description and in text - tere are no comments to microstructure visible on the figures!

page 5, line 141 - fig. 3 - the X-ray diffractions patterns have poor quality, especially the a) bottom fig. In such situation there is difficult to compare and interpretate results

Author Response

Dr. Yvonna Jiraskova

Institute of Physics of Materials,

Academy of Sciences of the Czech Republic,

Zizkova 22, 616 62 Brno

Czech Republic         

February 19, 2019

We thank Reviewer for his/her comments leading to improvement of our manuscript. We have accepted all of them.

All changes are marked by red colour in the revised version of the manuscript and explaining answers and information are added here:

Reply to Reviewer 2

Reviewer

page 4, line 128 - fig. 2. - the SEM micrographs should be evaluated, both - on the pictures - there are not any description and in text - tere are no comments to microstructure visible on the figures!

Answer

The text describing figure 2 (figure 3 in the corrected manuscript) was slightly changed as follows: 

3.1. SEM – Morphology and chemical composition

The surface morphology of the arc-melted samples (left-DS, right-DA) is seen Fig. 3aThe wheel-side morphologies of the ribbon-type samples of both compositions (RS, RA) are shown in Fig. 3b and morphology details in Fig. 3c. The same way is used for morphology imaging of the air-sides of both ribbon-type samples in Figs. 3d, e. The EDX chemical analysis taken from an area of about (1 x 1) mm2 resulted in good agreement with nominal composition mainly at the AM disc samples (DS, DA) where also the homogeneity of the chemical composition was satisfactory. This is reflected in a small dispersion of measured values in Table 1. Slightly different chemical compositions between wheel and air sides were found at ribbon samples (RS, RA). The largest local inhomogeneities of chemical composition were detected at the wheel side of the Fe2MnSi sample (RS) (Fig. 3b, left column). Point EDX analyses split into three groups: darker square objects yielded higher content of Mn at the expense of Fe (38.3±0.4 at.% Fe, 33.9±0.6 at.% Mn and 27.8±0.1 at.% Si), smaller bright objects yielded even higher content of Mn (in at.%: 26 Fe, 40 Mn, 34 Si), and the majority gray matrix yielded 45.3±0.1 Fe, 27.3±0.1 Mn, 27.4±0.1 Si (at.%) close to the nominal composition. All formations are well visible in Fig. 3c, left. Contrary to this situation, no similar inhomogeneities were observed at the wheel side of the RA sample (Fig. 3c and d, right column) even if the surface morphology was very similar. The large grains of dimension ranging between 250 μm and 300 μm were observed at samples prepared by AM (DS, DA – Fig. 3a) while the grains of both samples prepared by PFC were at least two orders of magnitude smaller in agreement with the higher cooling rate used at this technology. They are in detail seen in Fig. 3e.

 …

Figure 3. SEM micrographs of the Fe2MnSi (left panel) and Fe2MnAl (right panel) samples prepared by arc melting: DS, DA (a) and by planar flow casting - wheel side: RS, RA (b, c), air side: RS, RA (d, e). Details are described in the text.

Reviewer

page 5, line 141 - fig. 3 - the X-ray diffractions patterns have poor quality, especially the a) bottom fig. In such situation there is difficult to compare and interpretate results

Answer

Figure 3 (figure 4 in the corrected manuscript) was improved and we hope that its quality is now satisfactory.

Figure 4. X-ray diffraction patterns for Fe2MnSi (a) and Fe2MnAl (b) Heusler alloys prepared by arc melting (DS, DA) and planar flow casting (RS, RA).
